# Disasters and Society: Comparing the Shang and Mycenaean Response to Natural Phenomena through Text and Archaeology

Alexander Jan Dimitris Westra [1,2,3,*], Changhong Miao [1], Ioannis Liritzis [1] and Manolis Stefanakis [2]

1 Laboratory of Yellow River Cultural Heritage, Key Research Institute of Yellow River Civilization and Sustainable Development & Collaborative Innovation Center on Yellow River Civilization, Henan University, Kaifeng 475001, China; chhmiao@henu.edu.cn (C.M.); liritzis@henu.edu.cn (I.L.)

2 Department of Mediterranean Studies, University of the Aegean, 81100 Mytilene, Greece; stefanakis@aegean.gr

3 Chercheur Associé au Laboratoire ArScAn | Archéologie de l'Asie Centrale-UMR 7041, Université Paris Nanterre, Ministère de la Culture, 92000 Paris, France

* Correspondence: westra@henu.edu.cn

**Abstract:** Disasters do and have happened throughout human existence. Their traces are found in the environmental record, archaeological evidence, and historical chronicles. Societal responses to these events vary and depend on ecological and cultural constraints and opportunities. These elements are being discovered more and more on a global scale. When looking at disasters in antiquity, restoring the environmental and geographical context on both the macro- and microscale is necessary. The relationships between global climatic processes and microgeographical approaches ought to be understood by examining detailed societal strategies conceived in response to threatening natural phenomena. Architectural designs, human geography, political geography, technological artefacts, and textual testimony are linked to a society's inherited and real sense of natural threats, such as floods, earthquakes, fires, diseases, etc. The Shang and Mycenaean cultures are prime examples, among others, of Bronze Age societies with distinctive geographical, environmental, and cultural features and structures that defined their attitudes and responses to dangerous natural phenomena, such as floods, earthquakes, landslides, and drought. By leaning on two well-documented societies with little to no apparent similarities in environmental and cultural aspects and no credible evidence of contact, diffusion, or exchange, we can examine them free of the onus of diffused intangible and tangible cultural features. Even though some evidence of long-distance networks in the Bronze Age exists, they presumable had no impact on local adaptive strategies. The Aegean Sea and Yellow River cultural landscapes share many similarities and dissimilarities and vast territorial and cultural expansions. They have an apparent contemporaneity, and both recede and collapse at about the same time. Thus, through the microgeography of a few select Shang and Mycenaean sites and their relevant environmental, archaeological, and historical contexts, and through environmental effects on a global scale, we may understand chain events of scattered human societal changes, collapses, and revolutions on a structural level.

**Keywords:** cultural; tangible; Yellow River; catastrophic; religion; climatic; dynasty; flood; Shang; Mycenae; myth; history

## 1. Introduction

The manner of human interaction within a geographical area is determined by a series of cultural processes where the very geography of the location limits the evolution and layers of cultural values in each environmental niche. The layers of effects of structural geology, topography, climate, vegetation, hydrology on the settlements, economies, ideologies, myths, and cultures of the Aegean Sea and Yellow River regions must be included comprehensively. The Mycenaeans and Shang dealt with rivers, wetlands, dry areas, lack and excess of water, unpredictable rainfall and soil erosion, coastal environments with

the background of change and fluctuations in sea level, alluvial, and river systems, and other physical phenomena tied to the exploitation of an ecological niche. However, the idea of a unified, homogeneous response to individual environmental processes and common economic strategies would be misplaced.

Natural disasters are part of an ecological niche and include droughts, plagues, storms, volcanic eruptions, landslides, meteor impact and more [1–4]. They are perennial threats to any society whose traces must be sought in the environmental, archaeological, and historical records. Archaeology and environmental archaeology recognise changes in cultural and ecological systems. It is worth noting that in a period of environmental stability and conservation, the continuous habitation of an area for centuries is equally important to recognising changes in vegetation systems, erosion phases, and other natural phenomena [5] (Figure 1). Discussions like the present one link our contemporary ecological challenges and forthcoming disasters. They shine light on the successes and failures in the management of space, the insights and misunderstandings regarding the processes of nature, and the calamities that have consistently beset organised human life, a hard lesson humanity is seemingly obstinately unable to learn, even in the face of incipient and ongoing environmental disasters.

This paper aims to demonstrate some of the key discussions revolving around natural disasters witnessed in environmental data, archaeological evidence, epigraphic evidence, and historical texts. It focuses on primary discussions regarding the Shang and Mycenaean cultures and associated natural disasters. The paper develops from intertwined sections concerning environmental archaeology and natural disasters, physical and textual evidence of destructions, mythical destructions, and disaster management, prehistoric disaster management involving technology and mythology, natural disasters that led to collapse, archaeoseismicity, which is a catalyst for collapse and resilience, the case for China coupled with protoscientific applications, religious world-views, and cosmology supported by epigraphic evidence, texts, and ideas from later periods (Table 1).

**Table 1.** Table showing basic chronological comparison of Greece and China (dates from Bintliff, 2012; Underhill, 2013) [6,7].

| Greece | | China | |
|---|---|---|---|
| ca. 3200 BC | Early Bronze Age (EBA) | ca. 2550–ca. 1950 BC | Longshan Culture |
| ca. 2000/1900–ca. 1800/1700 BC | Middle Cycladic/Middle Helladic | ca. 2300–1500 BC | Qijia Culture |
| ca. 1800/1700–ca. 1500 BC | Late Cycladic 1/Late Helladic 1 | ca. 1750–ca. 1200 BC | Sanxingdui Culture |
| ca. 1500–ca. 1400 BC | Late Cycladic 2/Late Helladic 2 | ca. 1800–ca. 1450 BC | Yueshi Culture |
| ca. 1400–ca. 1250/1200 BC | Late Helladic 3A-B | ca. 1850–ca. 1550 BC | Erlitou Culture |
| ca. 1250/1200–ca. 1050 BC | Late Helladic 3C | ca. 1600–ca. 1400 BC | Erligang Culture/ Early Shang Culture |
| | | ca. 1400–ca. 1250 BC | Huanbei Culture/ Middle Shang Culture |
| | | ca. 1250–1046 BC | Yinxu Culture/ Late Shang Culture |
| | | 1046–771 BC | Wesstern Zhou Culture |

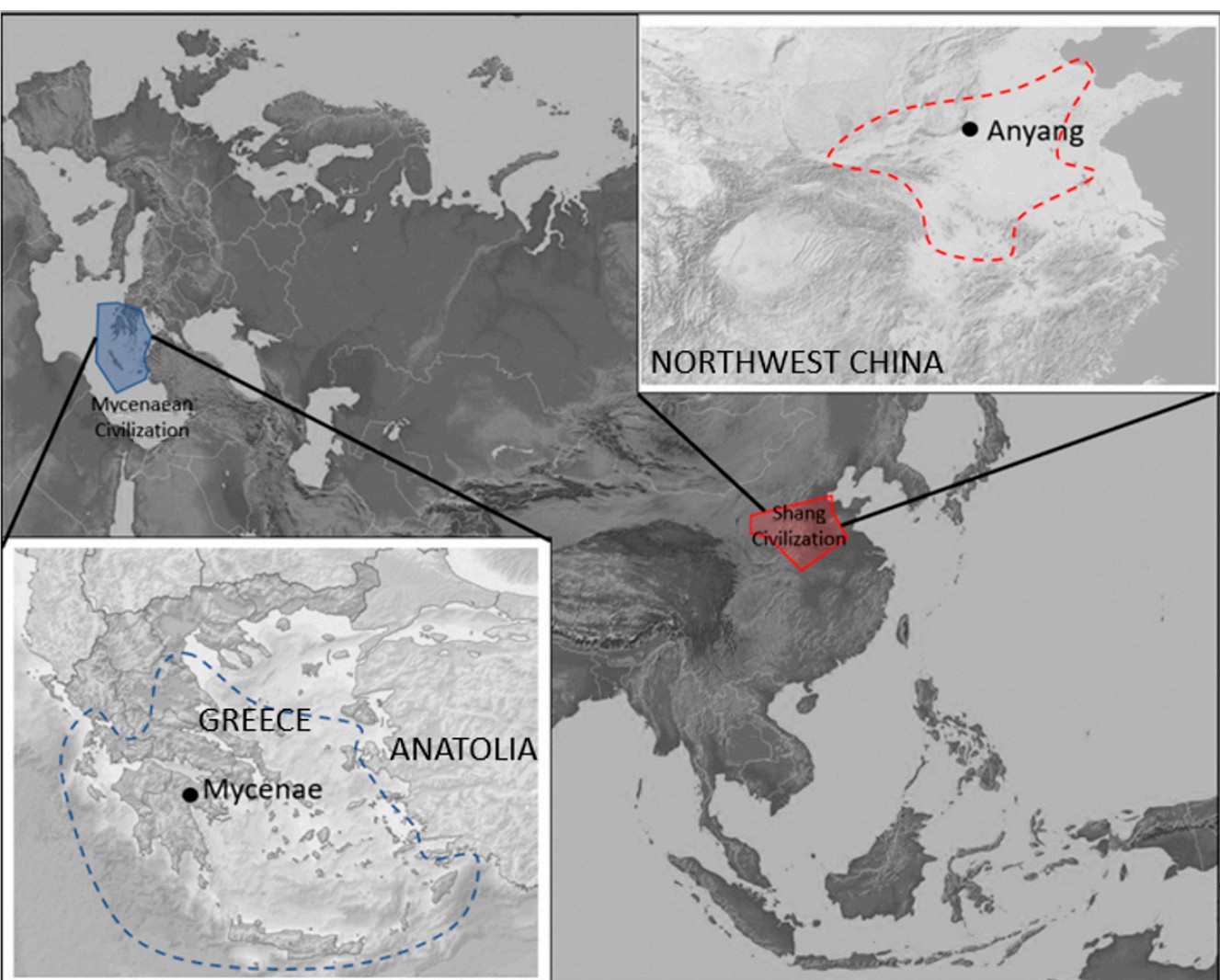

**Figure 1.** Approximate Mycenaean and Shang territories (Based on [8,9] (basemap free source: https://geology.com/world/asia-physical-map.shtml (access on 10 June 2022))).

## 2. Disasters through the Environment and Archaeology

### 2.1. Environmental Archaeology and Natural Disasters

Historical ecology, related to functional anthropology, provides archaeologists with concepts for evaluating cultural evolution [4]. Within cultural zones, technology and life-ways are associated and developed with their environmental context. By extension, the decline of a civilisation can be understood as an ecological disaster caused partly by the misuse of resources and catastrophic natural phenomena. The direct correlation of catastrophic events affecting prehistoric or ancient societies, in this case, the Late Bronze Age Aegean and the middle reach of the Yellow River, with the destruction of the Mycenaean or Shang civilizations, is widely discussed [10,11].

How and who applied environmental knowledge is a crucial question, as the participation of people in environmental change is self-evident and dynamic. Handling and responding to local environments reveal timeless cultures in specific landscape contexts. Environmental processes test social endurance: drought affects the supply of food, ideological and ritual mechanisms are developed to dampen the impact, and a social memory of disasters is created, which results in questioning the ability of the ruling class to appeal to a deity to mitigate threats from the natural world. The practical problem-solving of the Shang and Mycenaeans demonstrates the empirical approach to effectively minimising

the effects of earthquakes, floods, and other natural threats with different material and conceptual toolsets, as well as potential societal backlashes.

Distinguishing in the collapse of states and political units, the collapse of civilizations and 'Great Traditions' is not a straightforward matter. [12]. The terms 'collapse' and 'fall' are misleading, because the fragmentation of empires and states into smaller political units does not necessarily translate into a reduction in complexity [10], although in archaeology, it means a reduction in cultural and political complexity. A political collapse, i.e., the collapse or weakening of the state (political hegemony), incurs a chain reaction felt by different strata of society and is visible through archaeological evidence. However, cultures are distinct constellations of material and immaterial phenomena. Collapse does not necessarily lead to an all-out loss of complexity, customs, or the disappearance of the population [10]. Collapse on such a large scale paints historical and ecological collapses with broad strokes and reduces the lived experiences of communities as irrelevant to the continuity of larger abstract entities, such as culture, civilisation, or nations.

*2.2. Physical and Textual Evidence of Destructions*

For the causes of the Late Bronze Age collapse across the eastern Mediterranean, which spelt the end of the Mycenaean civilisation, a wide array of explanations have been proffered [13–17]. However, linking specific evidence of floods, earthquakes, or volcanic eruptions with prehistoric and legendary records remains complex and contentious, as the impacts and effects of disasters are not well understood [18]. For instance, whether the volcanic eruption on Santorini led to the sudden or gradual collapse of the Minoan civilization has been much debated since Marinatos [19]. Due to the reduced solar radiation and temperatures, the eruption's effects may have had local, regional, and global implications [20] The lack of precise chronologies rends the broader analysis of the relationship between a geological event or catastrophic natural phenomena and broader social and cultural implications hypothetical. For the eruption of the volcano of Thera/Santorini [21,22], the proposed date is circa 1620 BC. A different constellation of natural and political events surround the Thera eruption: a specific catastrophe occur at a precise point in time whose wider environmental and historical consequences are thoroughly investigated. Despite the well-documented effects across the Aegean and Eastern Mediterranean and an increasingly higher chronological resolution, the cause(s) of the Late Bronze Age collapse is unclear and its link to it yet to be reported [23,24].

The Yu and the "Great Flood" myth hypothesis states that an earthquake burst a dam in the Tibetan Plateau, resulting in the cataclysmic flood of the Yellow River basin area circa 2000 BCE, and has been increasingly debated since Wu et al. [11]. The scale and timing of the flood suggests some correspondence with the mythical "Great Flood" of the Yellow River and taming of the waters by the legendary figure of Yu. The story of the hero Yu dredging the water and earning himself the divine right to rule marks the establishment of the legendary Xia dynasty, according to the Shujing (Book of Documents) and the *Shiji* (Records of the Grand Historian Sima Qian). In the traditional Chinese historiographical and archaeological context, moral weight is given to the emergence of the Chinese civilisation from the cataclysmic floods and cultures and societies. Whether natural disasters are a force or catalyst for civilisational destruction, collapse, renewal, and emergence is a historiographical, philosophical, and theological schema and quandary that attributes judgement on ancient destructive natural phenomena. Contrarily to the value judgement placed on terrifying natural phenomena, the environmental and archaeological facts inform us of patterns, changes, shifts, and fluctuations in a scientific setting that ought to be reasonably and relatively void of socio-historical judgement and moralization. The comparison between the interpretations of the cases of the Thera eruption and the 2000 BCE flood, coinciding with the Xia dynasty, the Yellow River flood reveals more about the inherited historiographical, sociological, and disciplinary differences between the Archaeology of Greece and the Archaeology of China than about the prehistoric civilisational mechanisms of collapse or emergence (Figure 2).

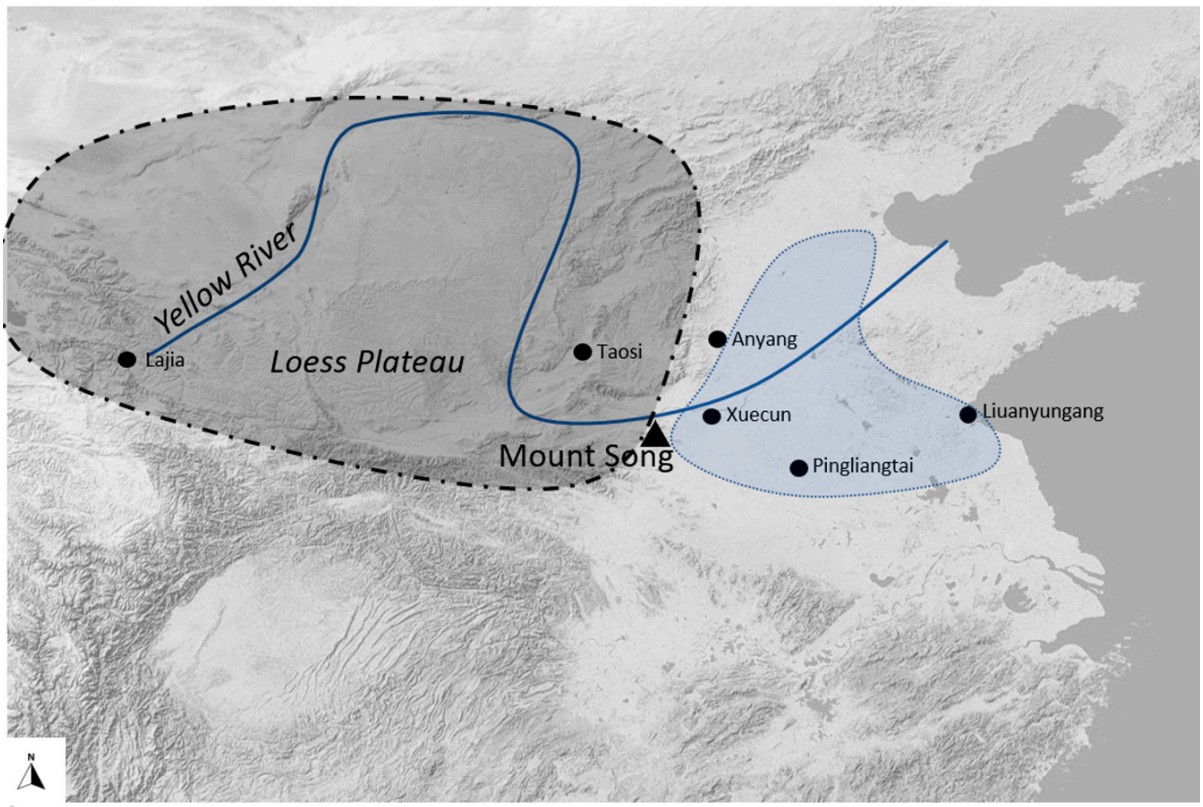

**Figure 2.** Map showing sites mentioned in the text and associated environmental features around the Yellow River basin such as Mount Song (triangle) (based on the flood dynamics described in [23], under a Creative Commons Attribution License). (basemap: free from Tom Patterson, US National Park Service Natural Earth), distance Taosi-Anyang = 256 km.

## 3. Early Cultural Engagement with Disasters

*3.1. Mythical Destructions and Disaster Management*

The experience of earthquakes, volcanic eruptions, and floods is traumatic and incites interest and disquiet in societies. Contemporary analyses and interpretations of past, current, and future destructive natural phenomena follow the historical schemas of ancient historical, philosophical, and theological treatises that considered conflagrations, cataclysms, inundations, and calamities with eschatological claims, moralising virtues, and renewal or renaissance stories, etc. [24]. In archaeology, doomsday is an old-fashioned way of thinking recently reinstated in the face of our contemporary degrading climate [18]. Some search for mythical and legendary catastrophic events as [25] true accounts and historical explanations are affirmed for the rupture and transformations of ancient civilisations, (i.e., Euhemerism). for example, the Biblical Floods, the destruction of Atlantis. and the Yu and Great Flood myth. The flooding of the Yellow River is a common and measurable phenomenon through various methods [26]. The evolution of its course and delta has resulted in large-scale floods in different periods, including the Longshan period [27,28]. One can take issue with the standpoint that considers the Yu and the Great Flood myth a historical *fait accompli* dated to circa 2000 BC (for example, Wang, 2005 [29]) However, the relevant figures of King Yao, Gun, and Yu are not proven historical personages, significant water-control activities worthy of such legend are unavailable, greater water management

capabilities of other cultures within the Chinese world are not considered, [30], and other rivers in within the floodplains [31] are omitted. Thus, there are severe scientific issues in taking the historicity of China's Yu and the Great Flood myth as a historical fact.

Such claims are circular in reasoning and controversial in their historical claims and moralisation. Other Euhemeristic practices are found in the minor and nebulous fields of geomythology [32] and archaeomythology [33]. In archaeology, such concerns are found in the long-standing debates concerning the historicity of the Homeric epics or the Xia and Shang dynasties, known as the "Homeric Question" and the "Doubting Antiquity School". Investigating prehistoric and historic societal responses to natural disasters can be fraught with biases and masked by the scientific veneer of environmental science, the venerability of ancient sources, and recent-to-contemporary sociocultural, ideological, and political implications and concerns. Archaeology emerged from an imperialist and colonial perspective and autochthonous (cultural, nationalist) perspectives that can influence contemporary inquiries. No field of study or discipline has emerged from a socio-cultural and ideological vacuum. A thorough comparative archaeological study necessitates a close review of historical and archaeological literature to bring to the surface some of the historical, cultural, and ideological biases surrounding a Sino-Hellenic study [34,35], which is beyond the scope of this paper.

*3.2. Prehistoric Disaster Management: Between Technology and Mythology*

How did the prehistoric Xia and Shang, and Minoan Mycenaeans engage with their environments? A utilitarian axiomatic claim sees a distinction between our contemporaries and our prehistoric ancestors that lies in tangible and intangible 'tools'. Yet, a prehistoric mind is not inferior (or superior) to ours [36] (paraphrased: 'prehistoric tools, not prehistoric people'). This implies that prehistoric technological developments, infrastructures, and state systems, as well as myths, cults, and religious and literary traditions, are part of a society's practical toolset it may call upon to predict, prevent, or mitigate the effects of inevitably forthcoming natural disasters. Reconstructing these 'toolsets' may render intelligible the 'holy' (spiritual-metaphysical) and the practical, or 'profane', processes (knowledge, materials) that lead to a society's eco-political successes and failures. Historical ecology evaluates environmental expertise and its involvement with the landscape and the environment of the community. The adaptation strategies of groups and developed technologies and lifestyles are not by default repeated in other similar eco-cultural settings [4]. Although Mediterranean cultures have common elements in landscape management, their strategies depend on various timeless historical, cultural, and economic processes. Anti-seismic structures, for example, the experiential knowledge of the Minoans in rebuilding their habitat and mitigation strategies against seismic damage, include building on the bedrock, using wooden frames, posts, and cross-beams set on stone socles, and mud-brick walls. Mud-brick is a global phenomenon, so it cannot be considered a Mediterranean anti-seismic technology. Instead, it sits within a wide array of tools available to each group of prehistoric people.

Technology features in myths, and the archaeological material reflects the visionary ambitions of the Shang and Mycenaean technophiles and the fundamental importance of technology [37] for the 'clans' of the middle reaches of the Yellow River and the Aegean Sea. The legendary achievements of conquest, trade, and arts relied on a fundament of successful agricultural practices. Despite the territorial expansion, vassalization and exchange and diplomacy within and outside the Mycenaean or Shang supremacies, each city and palace relied on locally produced agriculture for wealth and power. Having progressed beyond the irregular Neolithic agricultural practices, the dynastic clans of the Bronze Age, now firmly implanted permanently within palaces and citadels, sought to administer their 'estates'. Hence, reclaiming soils from the water, diverting river systems, draining valleys, irrigating plains, flood control, and storing water became increasingly advanced, efficient, and more significant in scale.

Military might is expressed through the monumental tombs of the citadel at Mycenae, art and material, the extensive cultural expansion across the Aegean, the conflicts with neighbouring larger empires, and the advanced architecture and engineering. Altogether, they testify to the dynamic and aggressive growth of the Mycenaeans [38]. Linear B supports the idea of a centralised administration, but leaves unclear the nature of the Mycenaean state as a kingdom, confederacy, or other. It is mostly discussed through the lens of Homeric epics [39,40]. The rich monumental tombs at Anyang, the burials of mass human sacrifices, large and complex military campaigns, coordinated garrisons, and territorial control, and the incredible and complex cities and productions point to the stratified society and terrifying power of the Shang dynasty. Yet, the oracular literature suggests that the territorial disparity and military might be propped up by allies and subordinates, which sent the ruler on permanent diplomatic missions [41]. More detailed accounts of the Shang state are available elsewhere [6,41–43].

In the archaeological literature, the conscious and ingenious water management capabilities of the Aegean and the Yellow River Bronze Age civilisations are studied to a greater extent than anti-seismic knowledge and engineering. The Greek god Poseidon—in Linear B tablets the 'Earthshaker' (Linear B: e-ne-si-da-o-ne) in Knossos [44]—is evidence of the sociocultural integration of the recurring destructive natural phenomenon into the Minoan and Mycenaean world-view and their associated cultic practices. The building techniques of the Mycenaeans, such as wooden framework wall construction, are found at Mycenae, Tiryns Thebes, and Knossos [45]. Anti-seismic construction techniques are also well attested by the Minoans, with foundations built on bedrock, wall frameworks, symmetrical plans, etc., that could withstand ordinary and extraordinary seismic events [46]. Minoans deployed anti-seismic construction methods that improved over time and extended to monumental buildings and private residences.

## 4. Disaster as Factor for Human Evolution

### 4.1. Natural Disaster to Collapse?

Anti-seismic measures are contained within the knowledge and material capabilities of the Shang and Mycenaeans and include both profound and far-reaching understanding and enduring misunderstandings about the inner workings of nature and physics. The Bronze Age societies of the Aegean and Yellow River regions suffered several disasters, leading to the loss of life and material and habitable or cultivable space. Yet, their way of life endured. Consequently, in investigating natural causes for a societal collapse, the question ought to be 'Why did this particular natural disaster prove to be the cause for collapse?'.

The Aegean region is marked by frequent seismic activity [47–49]. Growth in interdisciplinary work in Aegean archaeology is exemplified through archaeoseismology [50,51] and recently in Yellow River archaeology. Natural disasters in an archaeological context are social phenomena [52]. The societal impact of a prehistoric earthquake varies for each community. While they may cause a degree of local decline, or even collapse, they do not cause civilisational collapse [53]. A host of hypotheses exist that range from natural to societal causes, which do not need to be repeated (for a summary, see Middleton, 2020 [11]) (Figure 3).

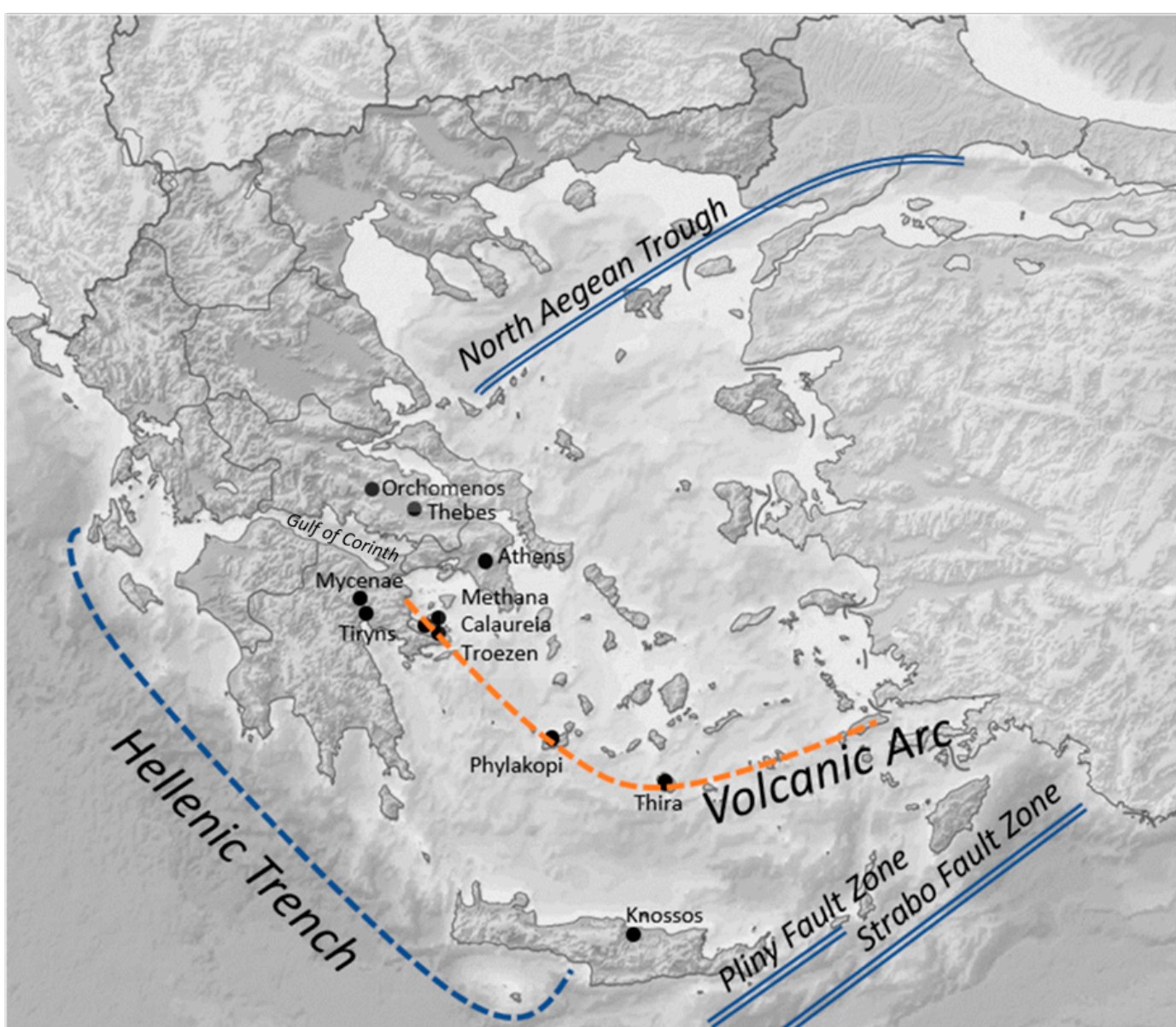

**Figure 3.** The Volcanic Arc and the Hellenic Trench in Greece with sites mentioned in the text. The Aegean Sea Plate (also called the Hellenic Plate or Aegean Plate) is a minor tectonic plate in the eastern Mediterranean Sea under southern Greece and far western Turkey. Its southern edge is a subduction zone south of Crete, where the African Plate is swept under the Aegean Sea Plate. Seismic and volcanic centers lie along the euro-african subduction zone. The Volcanic Arc (orange dashed line) of the northern Eurasian Plate is a divergent boundary responsible for the formation of the Gulf of Corinth. Note the northern Aegean Trench, the Hellenic Trench, and Pliny and Strabo fault zones in eastern Crete (based on [54,55], scale: Athens-Thebes = 205 km (Basemap: United States National Imagery and Mapping Agency data)).

Destruction of settlements happens, and earthquakes are not the preferred explanation. At first, Evans was reluctant to link the destruction of Knossos palace with seismic activity [51,56]. In addition, society's resilience against natural disasters also indicates human response to them [49,57]. The early Minoan palaces were destroyed between 1900 and 1700 BC by earthquakes [57–59]. The Minoan architectural discipline continuously evolved, incorporating new aesthetic features and practical anti-seismic features [50,60], such as building on bedrock, construction of smaller free-standing blocks, use of wooden

frames and cross-beams, façade projections and indentations [61], use of external friezes, lighter upper floors, and use of supportive pillars, among others [62].

Centuries after the Thera eruption, the destruction of the potentially overextended final Mycenaean palaces due to earthquakes or other events (i.e., invasion, Dorian invasion, return of the Heraclidae) do not signify the protracted end of the Mycenaean lifeways a couple of centuries later [63]. The Dorian invasion hypothesis has been rejected [64–66], and the notion of seismic storms causing the Late Bronze Age collapse (see: [57,59] is viewed with increasing scepticism [53]. The Mycenaean collapse is associated with the broader Late Bronze Age collapse of significant kingdoms and cities in the Eastern Mediterranean [67].

Critics reproach the indiscriminate explication of site abandonment or destruction to earthquakes when other factors are unknown [53] and deem it "neocatastrophism" [68,69]. Contrarily, the earthquake hypothesis is preferred, for the sake of parsimony, and stands against a myriad of unproven or unprovable complex hypotheses that can be affixed to the Mycenaean palatial collapse [57]. Subsequently, insufficient evidence and low temporal resolution have led to strained and circular explanations of earthquake-induced collapse or decline [70].

Mythological reports surround the Argolid area in the Peloponnese alleged natural disasters from river flooding [71], including other prehistoric geomythological issues of further deluges (Dardanus, Ogyges, Telchines). In addition, the fourth century BC earthquake and associated tsunami in the Corinthian gulf swallowed the ancient coastal city of Helike [72] and in the Levant [73].

A phenomenon as complex as *civilisational decline* or *collapse* is rife with reductionistic views and historiographical assumptions regarding what and how civilisational genesis, apex, nadir, and necrosis occur and are observable, or are useful as an interpretational paradigm. Adopting an interpretive framework of *civilisational decline* entails a series of complex phenomena of which catastrophic destruction is merely one aspect. In turn, discussing sudden collapse or gradual centuries-long decline also involves the disappearance of enough specific defined civilisational characteristics to render it 'other'. Cultural evolution is thus explained as a non-linear process [74].

### 4.2. Archaeoseismicity: A Catalyst for Collapse and Resilience

Disaster archaeology, in the case of the Mycenaean earthquake hypothesis, invokes several lines of questioning and observation to propose a conjunction of facts and events that risk forcing correlations into causation. An outcome of this discussion is the emergence of the multidisciplinary field of 'archaeoseismology' [53,75]. Archaeoseismology seeks to comprehensively and with interdisciplinarity determine the core seismic culture of an area [76]. Eyewitness accounts and later historical records have historical and sociological value in understanding emotive and pragmatic responses to the knowledge of an eventually forthcoming destructive natural phenomena and its aftermath and fallout. Historic seismicity allows us to extend the earthquake record considerably [77]. Paleoseismology examines the geomorphological and geological evidence for seismic fault activity [78], but its main weakness lies in providing a higher-resolution chronology for seismic events.

Indicators derived from destruction or damage of archaeological sites may provide a higher chronological resolution and the attraction of archaeoseismologists [50,79,80], but ascertaining whether victims, broken pots, and warped structures are the result of earthquakes or other physical pressures and processes is not an unambiguous observation to make [81]. In reality, archaeoseismological simulations of characteristic earthquake damage concluded that it could not be verified [82]. Numerous examples of seismic damage to Minoan and Mycenaean settlements and evidence of repair and reoccupation such as Knossos, Tiryns, and Mycenae exist. Yet, Hinzen et al.'s [83] assessment concluded that the evidence for seismic damage is often inconclusive and that earthquakes causing such widespread destruction to the end of the Mycenaean palatial system is unlikely.

### 4.3. Disaster Archaeology: The Case of China

The case of China, in comparison, offers different scales of magnitude and geography, and forces different perspectives on the notion of disaster. Earthquakes and floods have caused the demise of vast settled areas. Yet, a coherent culture continued to exist despite the calamities. On February 2nd 1556, an earthquake in Shanxi, Henan, and Shaanxi provinces resulted in the death of circa 800,000 people [84]. China's high population density can lead to tragedies of enormous proportions. However, determining this level of disaster in a prehistoric archaeological context is elusive. Archaeoenvironmental evidence regarding palaeoearthquakes in Xia- and Shang-era China is growing. For example, recent work in Henan Province revealed two faults, two grabens, and ground fissures, indicating a palaeoearthquake at Xuecun. The cultural layers and AMS 14 C from the ash pit date from the palaeoearthquake between 1520 and 1260 BC [85].

The buildings were built with regularly spaced timber posts, strengthened by horizontal cross-beams. The lack of nails and the use of interlock with mortises and tenons gave the building more flexibility and resistance to earthquakes. Earthquakes can also cause other natural disasters, such as floods. Architectural heritages such as *rammed earth* techniques appear independently in China and the Mediterranean regions, and constantly evolve [86]. In China, it is known as *hangtu* and in English as *cob* and is a widespread, simple, and earthquake-resistant building material found along the alluvial plains of north China since the Neolithic Longshan period (c. 3000–1900 BC) [87–89] at sites such as Pingliangtai [90], Lianyungang, Jiangsu, Taosi, and much more [91] as well as Bronze Age sites, such as Erlitou, Longwan, Anyang-Yinxu [92], and later historical periods (Warring States: 475–221 BC) [93]. The *rammed earth* structures (*hangtu*, *cob*, adobe, etc.) are known for their anti-seismic qualities, but whether that was consciously done by the first builder in China or elsewhere is unclear [94] (Figure 4).

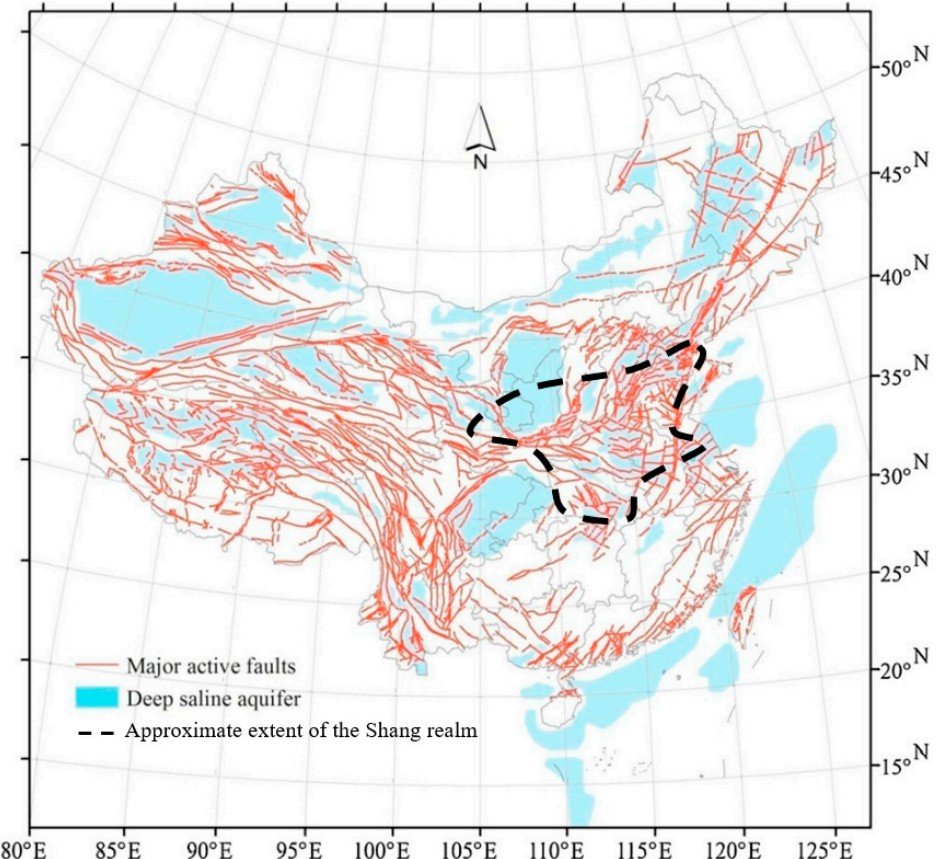

**Figure 4.** Present-day active seismic faults and deep saline aquifers related to the approximate extent of the Shang culture (dashed line) and beyond (based on source: [95], distances 5° = 555 km).

The Lajia Ruins, for example, are evidence of a natural disaster that destroyed human settlements. In the upper Yellow River, the destruction of the site of Lajia of the Qijia culture has been studied through various palaeoenvironmental means (micromorphology, OSL, paleosol). Around 2200–2000 BC, mudflows caused by rainstorms, flash floods, and earthquakes and the intensive exploitation of the landscape during the Neolithic resulted in the tragedy at Lajia 28, [96–100].

However, the prime suspect for the cause of the demise of the Shang has usually been sought and expressed through traditional historiography that derived from the accounts of Sima Qian (circa 145–86 BC). Nonetheless, the climatic shift in the northern central plains of China in the mid-Holocene is a clear marker of the far-reaching impact climatic change can have on modes of life. The historical connections between climate change and shifting frontiers between farmers and pastoralists of the northern Loess Plateau have received little attention [101]. Recent work has also proposed a connection between the collapse or decline of the Late Shang Dynasty and changing and deteriorating environments in the northern plains. Sedimentary data from the Loess Plateau indicates a sudden increase in aridity circa 1100 BC caused by the shifting maritime monsoon in favour of the continental monsoon. This climatic shift is also recorded in ice cores [102] and confirms its link to the aridity of northwest China [103]. Lake sediments from the Inner Mongolia Plateau show an abrupt reduction in water levels from circa 1100 BC to 400 BC [104,105]. Although not as sudden as an earthquake or flood, the relatively sharp increase in aridity would have destabilised the economy, production, and natural resources of the Shang Dynasty's traditional area of control, resulting in crop failures, livestock failures, mass migrations southwards, and encroachment by northern pastoralists. Altogether, increased aridity, leading to famines, instability, and uprising, could have set the scene for the collapse of the Shang Dynasty and the eventual Zhou takeover of the Shang realm [101]. The Yellow River culture of the Shang gradually expanded southwards in pursuit of copper and other resources, and before its demise, underwent a cultural revival [89,106–108].

Finally, understanding ancient seismicity varies according to each discipline's interest [53]. An exploration of the impact of disasters on the Mycenaean and Shang incorporates seismological, engineering, historical, and anthropological approaches [5]. Unlike sociological observations and conclusions, an archaeological investigation seeks to answer questions that relate to currents of the interests of the discipline, like chronology, and social adaptability, continuity, and discontinuity (Table 1). The placement of the citadel of Mycenae on an active limestone fault scarp (cf. [109], that could have split its cyclopean masonry in half gives us pause to consider the geological knowledge of the Mycenaeans [53] and more broadly the ecological and cultural understanding of the prehistoric Shang and Mycenaeans when faced with inevitable natural dangers. The recovery of Mycenaean and Shang knowledge is inferred from Bronze Age infrastructural projects, epigraphic evidence (Linear B, *Jiaguwen*), and later philosophical treatises. Natural phenomena were keenly observed and a core point of interest. Scientific and religious notions were developed within their own cultural and ideological frameworks and applied using tools, experiences, and knowledge. From later sources, earthquakes were not associated with tectonic plates, but with such causes as excessive Yin and trapped Yang, the winds, and the waters, which caused the world to shake. The overall cosmology of the Mycenaean and Shang visions of the world informed their cultic, religious, societal, and ideological approaches to dealing with natural disasters, while the practical applications visible in the archaeological record are derived from the experiential knowledge of materials and their properties.

## 5. Early Understanding of Disaster

### 5.1. Protoscience and Religion

Protoscientific applications, religious world-views, and cosmology can be linked to some extent, but they should not be confused. In other words, prayers, sacrifices, or appeals to Poseidon, Shang Di, or any deity or supernatural mover and properly built infrastructural works, such as dams, dykes, and canals, and disaster-prepared political

systems may both aim at avoiding and mitigating future disasters, but their approaches may have little in common.

The repeated occurrences of natural disasters have led the Chinese to compile probably the most reliable catalogue of them: the Guoyu Zhouyu (Historical Stories of States: Zhou), dating to the Spring and Autumn Period (Chunqiu) 770–476 BC. It reflects on how the Yellow River and other rivers drying up caused the collapse of the Shang [110]. Rivers drying up, droughts, famines, dust storms, and so on are mentioned in much of Chinese literature [101] and renders them plausible and the investigation into environmental causes for the collapse of the Shang Dynasties realm a valid hypothesis. The study of historical earthquakes in China is fertile ground, thanks to its extensive and accurate historical record. Since the Shang Dynasty, a variety of natural disasters have been recorded. The Catalogue of Chinese Earthquake, published in 1960, lists 585 historical earthquakes going as far back as 1189 BC. The Taiping Yulan, compiled by Li Fang and dating to the 10th century AD, lists 45 earthquakes between the 11th century BC and 618 AD [111,112]. Compared with the rest of the world, the Chinese records of historical earthquakes are the most complete [113]. The *Shiji* mentions how an earthquake in 780 BC, during the reign of King Yu of Zhou, interrupted the courses of three rivers.

> "In the second year of the reign of King Yu l (of Chou), the western province's three rivers shaken and their beds raised up, Poyang Fu said: 'The dynasty of the Chou is going to perish. It is necessary that the chhi of heaven and earth should not lose their order; if they overstep their order (it is because there When the Yang is hidden and cannot come forth, or when the Yin bars its way and it cannot rise up, then there is what we call an earthquake. Now we see that the three rivers have dried up by this shaking; it is because the Yang has lost its place and the Yin has overburdened it. When the Yang has lost its rank and finds itself (subordinate to) the Yin, the springs become closed, and when this has happened the kingdom must be lost. When water and earth are propitious the people make use of them, when they are not, the people are deprived of what they need. Formerly when the rivers I and Lo dried up, the dynasty of the Hsia perished. When the Ho dried up, the dynasty of the Shang perished.' Now the virtue of the Chou is in the same state as that of these dynasties was in their decline. The Chou will be ruined before ten years are out; so it is written in the cycle of numbers (Sima Qian, *Shiji*, Annals, 36 (Translation, [113] pp. 624–625)."

*5.2. Epigraphic Evidence*

Poseidon, the "Earthshaker" [*enesidao*], was prolifically worshipped in the Peloponnese. We turn to the prehistoric epigraphic textual evidence from the Linear B tablets and the *Jiaguwen* inscriptions (Oracle Bones). Natural phenomena, like earthquakes and floods, were rationalised by later Greeks and Chinese thinkers. Turning to proto-scientific explanations of natural phenomena, we must rely on contemporary epigraphic text, interpretation of archaeological evidence, and later philosophical treatises to reconstruct the Mycenaean and Shang modes of thinking.

Poseidon, the most commonly found name of a god in the proto-Greek Linear B tablets [114], is known as e-ne-si-da-o-ne ("Earthshaker") [115] from the 14th century BC Tablets in Knossos [114,116] and survives into the classical period as the god of the sea, earthquake, and tsunami [117] (see Figure 3).

The Linear B tablet mentioning Poseidon can also be linked to sanctuaries dedicated to Poseidon at Methana, Calaureia, and Troezen [118]. The cult of Poseidon and likely to a female counterpart by the Mycenaean is well established [119,120] and supported by the appearance of the name po-se-da-o (Tablet PY Es(-) 653 or PY Un(2) 6; also see Duhoux and Davis (2008) [121]), the cult at Phylakopi and the presence of a Hollow Psi figurine, and the pair of human figures driving the Methana chariot models [122]. This is also reinforced by the much later account of Diodorus (15.49.4), who describes the reverence paid by the Peloponnesians to their patron god, Poseidon.

> "That it was Poseidon's wrath that was wreaked upon these cities they allege that clear proofs are at hand: first, it is distinctly conceived that authority over earthquakes and floods belongs to this god, and also it is the ancient belief that the Peloponnese was an habitation of Poseidon; and this country is regarded as sacred in a way to Poseidon, and, speaking generally, all the cities in the Peloponnese pay honour to this god more than to any other of the immortals (15.49.4: *Diodorus Siculus*, tr. 1989 [123])."

Unlike the Mycenaeans, whose writing was mostly accounting, as it derives from the Mesopotamian tradition, the *Jiaguwen* reveals information relating to the divinatory and ancestral cult of the Shang [124,125], there is virtually no surviving secular use of writing [126].

In the Oracle Bones (*Jiaguwen*) inscriptions, commonplace divination is the encounter of some general 'disaster' or 'misfortune'. For instance, the character huò has been subject to much debate about the divination practices of the Shang.

The *Jiaguwen* texts used for divinations have mostly been found at Anyang, in Henan Province, at the site of the last Shang capital known as Yinxu [41,127,128]. After the Shang dynasty was overthrown, scapulomancy and plastromancy were still practised, but held much less importance in state affairs than in the Bronze Age [125,129,130]. The Oracle Bones is written in the form of Chinese that was maybe archaic for the late Shang kings, which would have been the 'language of the ancestors'. Within the corpus are mentioned sacrifices to the Yellow River (he) and the winds (Feng), suggesting an appeal to the spirits of natural elements [125]. The character yu or yuji is believed to mean 'apotropaic rite against disasters', or 'religious ritual performed to prevent and eliminate disasters' [131] such as illnesses, natural disasters, and crop failures:

> 10,152: Crack-making on xinyou, performs yu ritual for flood damage.
> 14,407: Crack-making on xinyou, divining: performs yu ritual for flood damage, and sheep are offered for rituals.
> 72: yu ritual . . . for Shangjia against disasters [131].

The Shang dynasty was established circa 1600 BC and inherited a host of religious ideas, scientific notions, and mythological traditions from previous and unknown cultures, as well as trends towards urbanisation and state formation. One should be careful not to project later historical accounts and modern understanding on such an archaic Bronze Age culture as was the Shang [41]. For a more detailed discussion of the religious landscape of early China, see [132,133]. An impression of the Shang religious beliefs could be summarised as natural phenomena considered as deities, such as the sun, the winds, the earth, the river, etc., whose connection with the Shang ancestral lineage is blurry. *He*, the river deity (spirit—*shen*, or force) and *Yue* (mountain) figure prominently in the Pantheon and are conventionally believed to refer to the Yellow River and *Songshan* (Mt. Song). Sacrifices were made to these *shen* (spirits) [41]. Lastly, the identity of *Di* has been debated. Responsible for the *huo* (catastrophe, disaster, ill omen), he is linked with weather and crops [134]. The *Shang* rulers appealed to *Di* to divine his intentions on projects, wars, and buildings [135].

### 5.3. Texts and Ideas from Later Periods

The growing interest in Sino-Hellenic comparative historical and philosophical studies [136,137] begins from the early historical periods of China and Greece. During the antiquity of Greece and China, several conceptions about natural disasters and their explanations are formulated. For instance, the *Shiji* quote refers to the Yin and Yang, a prevalent notion during the Han Dynasty that earthquakes could be foretold astrologically. In ideas found in the *I-Ching* (Book of Changes), earthquakes or thunders could be due to constrained Yang or an excess of Yin [113]. Whether such theories can be found with some approximation to periods before the eighth century BC is debatable [113]. The *Meteorologica*, reiterating Anaxagoras, stated that earthquakes were the result of water excess from the upper regions flowing into the hollows of the earth. Democritus considered that earth-

quakes were the result of soil saturated with water. Anaximenes proposed that seismic shocks result from masses of world falling in caverns. Aristotle put forward the notion of earthquakes being due to instability of the pneuma/vapour [138].

## 6. Conclusions

Ancient texts dating to periods several centuries after the disasters have sometimes been used to glean some clues and offer general remarks about prehistoric religion and beliefs regarding divination and natural disasters in Greece and China [130], in particular, the sociological implications of divinations as a political and rhetorical tool. Perhaps the comparison of Greek and Chinese mantic practices can be insightful for the archaeological interpretations of Shang and Mycenaean religion and sociological implications of natural events [138].

Yet, the remains of destruction and abandonment of once-thriving settlements are the staple of archaeological research. The causes that led to the sudden or gradual abandonment of settled sites and regions are not clear and must be examined. The catalogue of lost, destroyed, or abandoned community sites is large and incomplete. The fall of a once-thriving world contains the ingredients for many literary, political, religious, ideological, and overall cultural concerns and products. The *Bamboo Annals* and other works [139] describe earthquakes occurring during the time of the mythical emperor Fa (1831 BC) and the Gui emperor (1809 BC). However, their reliability and historicity remain doubtful [85]. Lastly, we may conclude that the Yu and the Great Flood parable, historicised in the Book of Documents (*Shiji*), which occurred in the time of the mythical King Yao, refers to a 'flood' (hongshui) from the mythical primordial sea or waters [140].

If we consider these myths, such as Yu the Great, Deucalion, Ogyges, and so on, as *living* fossils of human culture, then there is still no convincing methodical way of blending archaeological, historical, environmental, and legendary evidence. If there were no prehistoric people, just prehistoric tools, then the experiences of natural dangers and disasters and fears of a cataclysmic collapse of apocalyptical destruction were also woven into the fabric of their world-view, which was then expressed and addressed through practical and spiritual means. The Mycenaeans and the Shang were intelligent observers of their ecology. It can be assumed that no natural (or supernatural) phenomenon, regular or exceptional, was ignored. These occurrences grouped ecological phenomena and societal and spiritual beliefs, which blended into convictions about the triangular influence between deities on nature, nature, and humans (nature–humans–deities) and the ideological legitimacy of archaic rulership. The feats of the Mycenaean and Shang engineers and architects are impressive. Their resilience to floods, droughts, earthquakes, and other catastrophic natural phenomena is understood through practical, intellectual, emotive, ideological, and spiritual engagements with perennial threats.

**Author Contributions:** Conceptualization, I.L. and A.J.D.W.; methodology, I.L., A.J.D.W., M.S. and C.M.; software, I.L. and A.J.D.W.; resources, A.J.D.W., M.S., I.L. and C.M.; writing—original draft preparation, A.J.D.W. and I.L.; writing—review and editing, A.J.D.W., I.L. and C.M.; visualisation, A.J.D.W., I.L. and C.M.; supervision, I.L., M.S. and C.M.; project administration, I.L. and A.J.D.W. All authors have read and agreed to the published version of the manuscript.

**Funding:** This work was supported by the National Natural Science Foundation of China (Fund project host: Miao Changhong), grant number 42171186.

**Acknowledgments:** A.J.D.W expresses special thanks to the École Française d'Athènes 1-month grant for access to its library in Athens, Greece, which made this paper's completion possible. I.L. thanks the Laboratory of Yellow River Cultural Heritage, Key Research Institute of Yellow River Civilization and Sustainable Development & Collaborative Innovation Center on Yellow River Civilization, Henan University, China, for supporting the Sino-Hellenic Academic Project (www.huaxiahellas.com, last accessed 1 July 2022).

**Conflicts of Interest:** The authors declare no conflict of interest.

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
