# Peer review of "Disasters and Society: Comparing the Shang and Mycenaean Response to Natural Phenomena through Text and Archaeology"

_quaternary, doi:10.3390/quat5030033_

Round 1
Reviewer 1 Report
This is an innovative comparison, with nearly perfect matches in time periods but in totally separate parts of the world. But make it clear that you are focusing on earlier events (during Minoan, Xia times) and later societies (Mycenaeans, Shang). Add a table (or two) showing the cultures/chronology for the Aegean and China so that the audience understands the timing that is referred to in the text.
The maps (Figs. 1, 2, 3, 4) all need scales and north arrows. Fig. 4 should also have some labeling of where the fault lines are.
There are MANY minor corrections to be made, see the attached file. Make sure that "Thira" is changed to "Thera".
The references in particular need corrections/proofreading of both content and the consistency of format

Reviewer 2 Report
This is an interesting, but overly long paper. it needs to be made more succinct, there is far too much 'extra' material, very wordy and the meaning or relevance of the material is lost quite easily. I found myself constantly having to backtrack to understand what the point of various sentences or even paragraphs were supposed to focusing on. There are English-related issues that the editors will have to fix. I think there is space for a discussion about the relevance and debates of oral and textual histories in relation to natural disasters. I would also like to see additional headings and sub-headings to signpost where the discussion is heading to next. Overall, you need to make the paper sharper, snappier and easy to read. Try using shorter sentences and more active language please to get your point across more clearly.
Reviewer 3 Report
I recommend the article for publication, I have a few criticisms and recommendations. First I list the essential ones, then just recommendations for correction (for the authors).
1) At the beginning I was surprised by the lack of line spacing. How can I comment on the lines if they are missing? Can't the authors and editor keep track of this?
2) At the end of the introduction, the aims of the paper are not clearly and concisely defined.
3) The absence of AIMS also makes the text clumsy. It is difficult to orient oneself in it and the main point of the paper is lost.
4) Many foreign terms make it difficult to read. They can certainly be replaced appropriately.
5) The references at the end do not have a clear form.
Page 1 - Abstract:
to dangerous natural phenomena. - specify
Introduction:
Page 2: Natural disasters (climatic disasters, etc.) are part of an ecological niche and include droughts,
plagues, storms, volcanic eruptions, landslides, and more. - add citation
I suggest to use this book in introduction and discussion:
MONTGOMERY, David R. Dirt: the erosion of civilizations. Berkeley: University of California Press, ©2007. ix, 285 s. ISBN 978-0-520-25806-8.
All figures in the article miss citation of basemaps.
Page 3: Historical ecology, related to functional anthropology, – add citation
truction of the Mycenean or Shang Civilizations
is widely discussed and questioned. – add citation
Mediterranean, which spelt out the end of the Mycenaean civilisation, have been a hotly debated topic, and a wide array of explanations have been proffered (Middleton, 2020, Renfrew 1984). – add more citations, this part cites a lot Middleton, it is not enough
remains complex and contentious as the impacts and effects of disasters are not well understood. – add citation
Page 5: euhemeristic – use other word
Page 6: China’s Dayu flood myth – missing dot
Sino-Hellenic study. – add citation
by default repeated in other similar settings. – add citation
haphazard – use other word
Page 7: aggressive growth of the Mycenaeans. – add citation
through Homeric epics. – no dot
Page 9: genesis, apex, nadir, and necrosis – please explain
(Liritzis 2013). – add comma
Page 10: Shang-era China is growing. – add citation
China is prone to significant seismicity. – repeated information above
Page 11: (micromorphology, OSL, paleosol) – add citation
Page 12: social adaptability – remove space
discontinuity and so on. – remove so on, the same further
conflated – ???
Page 15: â–¡ divining – remove square
